# Effects of Neem (*Azadirachta indica*) Leaf Powder Supplementation on Rumen Fermentation, Feed Intake, Apparent Digestibility and Performance in Omani Sheep

**DOI:** 10.3390/ani12223146

**Published:** 2022-11-14

**Authors:** Hani M. El-Zaiat, Elshafie I. Elshafie, Waleed Al-Marzooqi, Kawakob Al Dughaishi

**Affiliations:** 1Department of Animal and Veterinary Sciences, College of Agricultural and Marine Sciences, Sultan Qaboos University, Al-Khoud 123 P.O. Box 34, Oman; 2Department of Animal and Fish Production, Faculty of Agriculture, Alexandria University, Aflaton St., El-Shatby, Alexandria P.O. Box 21545, Egypt; 3Ministry of Agriculture, Fisheries Wealth and Water Resources, Saal, Al-Khoud 123 P.O. Box 34, Oman

**Keywords:** growth, monensin, neem leaf, nitrogen utilization, ruminal characteristics, sheep

## Abstract

**Simple Summary:**

Phytogenic supplements can be used as natural feed additives through their potential effects on rumen fermentation and the growth performance of lambs. Compared to monensin (MON), dietary supplementation with neem leaf powder (NLP) could promote body weight gain with no detrimental effects on the ruminal fermentation profile, nutrient intake, or digestibility. The results of this study suggest that NLP could be used as a promising phytogenic supplement in sheep diets rather than synthetic growth promoters.

**Abstract:**

The objective of the present study was to evaluate the potential of the dietary addition of neem (*Azadirachta indica*) leaf powder (NLP) when compared to monensin (MON) on ruminal fermentation, feed intake, digestibility, and performance of growing lambs. Eighteen Omani lambs (22.8 ± 2.18 kg of body weight (BW)) were equally divided into three groups (*n* = 6 lambs/group) for 90 days. Animals were fed an *ad lib* basal diet consisting of Rhodes grass (*Chloris gayana*) hay (600 g/kg) and a concentrated mixture (400 g/kg) offered twice daily. Experimental treatments were control (basal diet without supplements); MON (control plus 35 mg/kg DM as a positive control); and NLP (control plus 40 g/kg DM). Lambs fed NLP had reduced ruminal ammonia nitrogen concentrations, protozoal counts, total volatile fatty acid, and blood urea nitrogen concentrations compared to the control. Compared to MON, lambs fed NLP had increased ruminal acetate and decreased propionate proportions. Inclusion of NLP in the diet increased blood total protein, globulin, and liver enzyme concentrations in comparison with the control, which was similar to MON. The lamb’s final BW and average BW gain were also increased with the NLP relative to the control. Further, adding NLP to the diet increased the digestibility of crude protein compared to the control diet. In conclusion, adding NLP to the diet with 40 g/kg DM could be used as a promising phytogenic supplement for growing lambs with no detrimental effects on the ruminal fermentation profile, nutrient intake, or digestibility.

## 1. Introduction

For decades, monensin (MON) has been among the most extensively and successfully utilized ionophores as feed additives in livestock diets to promote animal health and performance. Such MON has been distinguished as a growth promoter capable of altering rumen fermentation patterns, enhancing energy efficiency, and feed utilization [1]. Dietary supplementation with MON shifted ruminal volatile fatty acids (VFAs) profiles toward more propionate and less acetate, leading to a decrease in methanogenesis and ammonia concentrations through influencing feed degradability [1,2,3]. The use of ionophores in ruminant diets has been banned [4] owing to the excretion of residues in animal products (such as milk and meat) and the environment [5] and the risks that such residues may pose to the public’s health [6]. Therefore, and from a nutritional standpoint, using phytogenic feed supplements in ruminant diets instead of synthetic antibiotics are preferable to modifying the rumen and overcoming livestock production challenges [7]. The use of tropical tree forages grown in developing countries, such as the Neem tree (*Azadirachta indica*), can help alleviate feed scarcity challenges in terms of quantity and quality throughout the feeding during drought seasons [8,9]. Tropical trees of the *Meliaceae* family, particularly neem trees, are native to tropical and subtropical regions; however, they are also available in other countries throughout the world [10]. Neem trees contain appreciated levels of limonoids (tetranortriterpenoids) bioactive secondary metabolites known as *Azadirachtin*, and these bioactive agents [10] would modify rumen microbial fermentation and overall animal performance [11,12]. A recent in vivo study by Webb et al. [7] using the extract of neem leaf had no effect on the carcass characteristics and fatty acid composition of South African Mutton Merino lambs. Feeding of neem fruit biomass up to 5% has been reported to increase microbial protein production, however, it reduced ruminal microbial populations, ammonia nitrogen, and propionate in West African dwarf rams [13]. Additionally, the medicinal potency of neem leaves [11,14] as antihelminthic therapies [15,16,17] has been investigated. However, in vivo studies focusing on comparing the efficacy of feeding whole neem leaf powder (NLP) as a natural feed supplement with MON on ruminal fermentation profiles and growth performance have not been fully investigated in growing lambs. We hypothesized that lambs fed with NLP would have ruminal fermentation patterns, blood profiles, nutrient digestibility, and performance similar to lambs fed with a MON diet. Therefore, the objectives of this study were to evaluate the efficacy of dietary addition with NLP on ruminal fermentation and blood variables, feed intake, nutrient digestibility, and growth performance of Omani lambs.

## 2. Materials and Methods

The experiment was conducted at the Sheep Research Unit located at the Agricultural Experiment Station of Sultan Qaboos University (Approved code number: RF/AGR/ANVS20/03; Approval date: 1 September 2020). All experimental procedures were carried out in accordance with the experimental unit policy on animal welfare, and the requirements of the procedures involving animals and their care were conducted in conformity with international laws and policies (EEC Council directives 86/609, OJL 358, 1 December, 12, 1987; NIH Guide for the Care and Use of Laboratory Animals, NIH Publications No. 85–23, 1985) at Sultan Qaboos University.

### 2.1. Experimental Feed Supplements and Preparations

Fresh neem leaves were collected from trees at Sultan Qaboos University’s Agricultural Experiment Station (23°59′89.8″ N, 58°16′25.6″ E). The fresh leaves were separated from the stalks, and thoroughly washed to remove dirt and other unwanted substances, then air-dried under the shade at an average temperature of 28.5 °C for one week. Subsequently, the dried leaves were ground into a powder according to the modified method of Hossain et al. [18] using a hammer mill (Alvan Blanch, Chelworth, Malmesbury, UK). The NLP was stored at room temperature in a large plastic container until needed.

### 2.2. Animals, Experimental Design, and Diets

Eighteen Omani male lambs [22.8 ± 2.18 kg of body weight (BW)] were equally divided into 3 groups of 6 lambs in a completely randomized design. Lambs were housed in individual pens located in a covered shed with a cement floor equipped with one drinker and feeder troughs. Animals were allocated to one of three experimental groups (one lamb/pen per treatment). Dietary treatments were control (basal diet without supplements), control diet supplemented with monensin sodium (MON, 35 mg/kg DM) as a positive control, or control diet supplemented with NLP (40 g/kg DM). The MON dose was chosen based on the manufacturer’s recommendation (Elanco Animal Health, Greenfield, IN, USA), and our previous studies [2,19]. A preliminary in vitro assay was carried out to examine the optimal level of NLP by using a semi-automatic gas production system. Four incremental doses of NLP were evaluated for ruminal fermentation characteristics, and feed degradability. Thus, based on our in vitro dose-response findings (data not shown), the dose of NLP was chosen for the current in vivo study. The chopped Rhodes (*Chloris gayana*) grass hay was mixed with the concentrate mixture to obtain the total mixed diet (60:40 ratio). The basal diet was formulated according to NRC [20] recommendations for the nutrient requirements of sheep to achieve a moderate average daily gain (ADG) of approximately 100–150 g/day. Lambs were fed their basal total mixed diet individually twice a day (900 h and 1600 h in equally sized portions), allowing 10% of the orts amount to be removed daily before the morning feeding with free access to water. The feed ingredients and chemical compositions of the experimental NLP, Rhodes grass hay, and basal diet are shown in Table 1. The experimental period was 105 days, which included 15 days for basal diet adaptation and 90 days for data and sample collection. Prior to morning feeding, the daily amounts of NLP and MON were weighed, and hand mixed with the basal diet to guarantee that all lambs received the exact experimental supplement amounts for each animal.

### 2.3. Feed Intake and Growth Performance

The amount of daily dry matter (DM) intake was calculated based on the difference between the amount of feed offered and the number of refusals. Animals’ live weight was taken individually using a digital scale on days 30, 60, and 90 of the experiment. The ADG was calculated as the slope of the linear regression of animal live weight on weighing time. Feed efficiency was calculated as ADG divided by DMI (g of ADG/g of DMI).

### 2.4. Apparent Nutrients Digestibility and Nitrogen Balance

After 90 days of the growing trial, all animals were moved to individual metabolism crates (1.0 × 1.5 m) for apparent digestibility and nitrogen utilization determination as described by Soltan et al. [19]. Individual refusals, feces, and urine were collected and measured daily for each animal before the morning feeding throughout the collection period (3 days of adaptation and 6 consecutive days of collection period). At the end of the sampling period, samples were pooled for each lamb and stored at −20 °C for later analysis. At the end of the digestibility trial, a fecal sample (100 g/kg, wet weight) of the total output was retained daily and dried at 60 °C for 72 h. Apparent nutrient digestibility was determined using the following equations: 100× (nutrient intake-fecal nutrient output)/nutrient intake. A daily 10% urine sample was collected and frozen at −20 °C until analysis for total nitrogen. The retained nitrogen was calculated by subtracting the excreted nitrogen (fecal nitrogen plus urine nitrogen) from the total nitrogen intake.

### 2.5. Rumen Fermentation Variables Measurements

Ruminal fluid (50 mL) samples were collected within 3 h from all animals after morning feeding on days 30, 60, and 90 of the experiment. Rumen samples were collected using a custom-designed stomach tube approximately 1 m in length 3 consecutive times in order to minimize saliva contamination and increase sample representativeness. Immediately after collection, the pH of ruminal fluid was determined using a digital portable pH meter (HANNA Instruments, HI microcomputer 9025, pH Meter, Smithfield, VA, USA). After recording ruminal pH, samples (5 mL of rumen fluid) were squeezed through four layers of cheesecloth in order to eliminate large feed particles. A rumen fluid subsample was centrifuged (SIGMA Laborzentrifugen, D-37520, Osterode am Harz, Germany) at 2000× *g* for 15 min at 4 °C, and the supernatant was collected and frozen at −20 °C before being analyzed for ruminal ammonia (NH_3_-N) concentration as described by Soltan et al. [18]. Subsequently, the supernatant was measured colorimetrically at 540 nm using a commercial lab test as described by Konitzer and Voigt [21].

Rumen filtrated subsamples were used for total volatile fatty acid (VFA) and individual VFA determinations. In brief, one mL of 25% metaphosphoric acid was thoroughly mixed into two mL of rumen fluid sample and left for 30 min to homogenize. Subsequently, samples were centrifuged (Eppendorf, 5702 R, Hamburg, Germany) at 15,000× *g* for 20 min at 4 °C. Approximately 1 mL of clear supernatant was collected into a GC vial and stored at a cold condition until analysis. The samples were analyzed using a gas chromatograph (GC, Agilent 6890 N) equipped with a flame ionization detector (FID) according to the method of El-Zaiat et al. [12]. 

A 2 mL aliquot subsample was mixed with 4 mL of methyl green-formalin-saline solution in a glass container at room temperature for a total protozoa count using the Neubauer improved bright-line counting chamber according to the protocol described by Dehority et al. [22].

### 2.6. Blood Biochemical Constituents

Blood samples were collected on days 30, 60, and 90 of the experiment before the morning feeding via the jugular vein from each animal. Blood serum was obtained using 7 mL vacutainer tubes (BD-Belliver Industrial Estate, Plymouth PL6 7BP UK) immediately after centrifugation at 2000× *g* for 15 min, and then stored at −80 °C until further biochemical variable analyses. For the blood hematological analyses, samples were collected using vacuum-glass (4.5 mL vacutainer) tubes (BD-Belliver Industrial Estate, Plymouth PL6 7BP UK) coated with K3 EDTA anticoagulant. Blood samples were examined for red blood cells (RBC), hemoglobin (Hb), white blood cells (WBC), packed cell volume (PCV), albumin, creatinine, urea-N, glucose, total bilirubin, glutamic oxaloacetic transaminase (GOT), glutamic pyruvic transaminase (GPT), alkaline phosphatase (ALP), and gamma-glutamyl transferase (GGT) were evaluated. Whole blood samples were analysed by the Abbott CELL-DYN 3700 blood analyzer (CELL-DYN 3700; Abbott Laboratories, Chicago, IL, USA) within two hours of blood sampling. 

### 2.7. Fecal Egg Count (FEC) Quantification

Individual fecal samples were collected at the same time as blood collections (on days 30, 60, and 90 of the trial) to quantify FEC by the modified McMaster technique [23]. In brief, 3 g of feces were weighed, and 30 mL of flotation solution (saturated sodium chloride) was added. Then the sample was homogenized and mixed. While stirring with a pipette, the liquid phase was sucked to fill both McMaster chambers, which were placed within the microscope stage; then, the total number was multiplied by 50 to obtain the total eggs per gram.

### 2.8. Laboratory Analyses

Basal diet, orts, and fecal samples were thawed and dried in a forced-air (GALLENKAMP, 300 plus series) oven at 60 °C for 48 h and then ground through a 1 mm screen using a mill (Tector; CEMOTEC, 1090 mill, Hillerød, Denmark). Ground basal diet and feces samples were chemically analyzed according to AOAC [24] for DM (ID method 934.01) and OM (ID method 942.05). Crude protein (CP, as 6.25 × N) was determined using (BUCHI KjelMaster K-375, Flawil, Switzerland) nitrogen/protein Analyzer (ID method 976.05). Ether extract (EE) was determined using petroleum ether by Soxhlet extraction of the dry sample (ID method 920.39). The concentration of neutral detergent fibre (NDF) was determined sequentially without heat stable-amylase using sodium sulphite and sodium lauryl sulphate as described by Van Soest et al. [25]. However, acid detergent fibre (ADF) was determined as described by Roberston and Van Soest [26] using cetyl trimethyl ammonium bromide (CTAB) and 1N H_2_SO_4_. The NDF and ADF fractions were not expressed exclusively from residual ash. Non-fiber carbohydrates (NFC, g/kg) were calculated as 1000 − (NDF + CP + EE + ash).

### 2.9. Statistical Analysis

The data were analyzed using JMP, version 9.0.2, by the SAS Institute. The continuous parameters measured were entered into a MIXED model analysis of variance for repeated measurements on days 30, 60, and 90 using the following model: Yij = μ + Ti + Eij where Yij = observations mean, μ = overall mean, Ti = treatment fixed effect (i = Control, MON or NLP), and Eij = residual error. Animals were nested in groups and were considered a random factor in the model. Normality was checked for individual factors, and transformation was conducted for the required parameters. Treatment differences were declared significant when *p* ≤ 0.05 and tendency was 0.05 < *p* ≤ 0.10 using the Tukey correction for multiple comparisons.

## 3. Results

### 3.1. Daily Feed Intake and Growth Performance

Daily DM intake was not affected by the inclusion of NLP compared to the control diet (Table 2). However, lambs supplemented with MON had lower daily DM intake (expressed in g/day; *p* < 0.001; and g/kg^0.75^; *p* = 0.002) in comparison to those fed NLP and control diets. In addition, final BW (kg; *p* = 0.029) and ADG (g/day; *p* = 0.014) were increased in lambs fed NLP compared to those fed the control diet but were similar to MON. Therefore, lambs fed NLP had no effect on feed efficiency compared with the control diet. However, supplementation of MON resulted in higher (*p* = 0.002) feed efficiency than those fed with the control or NLP diet. 

### 3.2. Intakes, Apparent Nutrients Digestibility, and Nitrogen Utilization

Intakes of DM, OM, CP, EE, NDF, and ADF were not affected between lambs fed with NLP and those fed the control diet (Table 3). Nevertheless, compared to NLP and control, lambs fed with the MON had decreased intakes of DM (*p* = 0.014), OM (*p* = 0.005), CP (*p* = 0.034), EE (*p* = 0.013), NDF (*p* = 0.024), and ADF (*p* = 0.043), expressed in g/day. Supplementation of NLP had no effect on DM, OM, or NDF digestibility compared to the control but decreased (*p* < 0.05) with MON. However, the addition of NLP increased the digestibility of CP (*p* = 0.038) when compared to the control diet. NLP supplementation did not change CP digestibility compared to MON. Adding NLP had no difference in nitrogen intake and fecal nitrogen excretion compared to the control group. Despite this, the addition of MON decreased dietary N intake (*p* = 0.005) and fecal N excretion (*p* = 0.042) compared to NLP and control diets.

### 3.3. Ruminal Fermentation Constituents

Lambs fed NLP showed no change in ruminal pH compared to those fed control (Table 4). In addition, adding NLP decreased ruminal NH_3_-N concentrations (*p* < 0.001), protozoal counts (*p* = 0.006), and total VFA concentrations (*p* < 0.001) compared to the control. However, the addition of NLP has no effect on ruminal acetate and propionate proportions compared to the control diet. Furthermore, when compared to MON, the addition of NLP increased ruminal acetate (*p* < 0.001) and decreased propionate (*p* = 0.017). The acetate-to-propionate ratio did not differ between lambs fed with NLP and control, but it decreased (*p* = 0.001) in lambs fed MON.

### 3.4. Blood Metabolites and Fecal Egg Count

Lambs fed NLP had a decreased PCV percentage (*p* = 0.009) compared to those fed the control diet (Table 5). However, this percentage was not affected between lambs fed NLP and MON diets. Blood total protein (*p* = 0.017) and globulin concentrations (*p* = 0.042) were higher in lambs fed NLP than in those fed the control diet, which was similar to those fed MON. Compared to the control, the blood urea-N concentration was lower (*p* = 0.032) in lambs fed NLP but was similar to those fed MON. Moreover, lambs fed NLP had no effect on total blood bilirubin concentration compared to the control, which was decreased (*p* = 0.021) with MON. Supplementation of NLP increased blood concentrations of GPT (*p* < 0.001), GOT (*p* < 0.001), and GGT (*p* = 0.022) compared to the control and MON groups. However, GOT and GGT concentrations did not differ between animals fed control and MON. Furthermore, blood ALP concentration did not change between lambs fed NLP and the control. However, this concentration was higher (*p* = 0.033) in lambs fed NLP compared to MON. In addition, lambs fed with NLP had no difference in total fecal egg count response to those fed control, but it increased (*p* < 0.001) in lambs fed MON.

## 4. Discussion

### 4.1. Daily Feed Intake and Growth Performance

The effectiveness of MON as a feed additive in improving livestock productivity and health has been well documented [27,28,29]. In the current study, dietary supplementation with NLP is preferable as, despite comparable daily DM intake, the ADG was higher (10%) than in the control diet. Similarly, though lambs fed NLP had a higher daily DM intake compared to those fed MON, the ADG was not increased; therefore, a reduction in feed efficiency resulted. These results agree with Jack et al. [30], who stated that adding neem fruit at 5% in the diet increased the ADG of West African dwarf rams as a result of better nutrient digestive efficiency. However, increased ADG and final BW in lambs fed NLP compared to those fed control did not respond with unaffected daily DM intake. These differences in ADG and final BW could likely be due to the observed differences in CP digestibility. Dietary supplementation with NLP achieved greater lambs’ BW gain, which is likely due to the role of phenolic compounds [18,31] to protect dietary protein from ruminal degradation, thereby enhancing the efficiency of dietary protein utilization. This result is possibly due to greater CP escape from rumen degradation, making more CP available to the abomasum and small intestine. The elimination of ruminal protozoa abundance resulted in an increase in the total bacterial community, thereby enhancing animal performance [32]. This finding was justified by greater CP digestibility. Similar performance results were obtained when goats were supplemented with sole neem leaves instead of concentrate mix and dry pigeon pea (*Cajanus cajan*), as reported by Dida et al. [8]. 

In addition, the increases in lambs’ ADG may possibly be associated with the increased blood total protein, suggesting more dietary protein is available to the animals [33]. To the best of our knowledge, this is the first study to highlight the effects of NLP as a phytogenic feed supplement on rumen fermentation, nutrient intake, digestibility, and growth in Omani growing lambs. From an economic perspective, the significant increase in feed efficiency when compared to the control demonstrates that lambs that received NLP could be finished in a brief feedlot system due to the greatest ADG, which is typically desirable for a substantial feedlot production system. However, the small number of experimental animals (six per group) used in this study may be the reason why no significant differences in feed intake were observed between NLP compared to the control diet.

### 4.2. Intakes, Nutrients Digestibility, and Nitrogen Utilization

Similar to the observed fermentation profile, no positive or negative effects were observed on the nutrient intakes of lambs fed NLP compared to the control, confirming the palatability of NLP. Therefore, feeding responses to NLP supplementation are deemed satisfactory in the current study since its bitter taste may negatively affect feed palatability and, consequently, DM intake. Our results coincide with those of Chandrawathani et al. [15], who found that the dietary addition of neem leaves was safe and acceptable with no adverse effects on daily feed consumption in sheep. A possible explanation for the similar nutrient intakes goes to lambs’ acclimation to neem leaves during the dry season, despite their bitter taste, as demonstrated by Adjorlolo et al. [34]. Compared to lambs fed the control diet, the nutrient intakes and digestibility of DM, OM, or NDF were not significantly affected by NLP supplementation which might suggest that the NLP used in this study did not impair the extent or rate of digestion. Supplementation of neem fruit biomass to ram’s diet has been shown to activate some ruminal microbes, as reported by Jack et al. [13], leading to increase intake and apparent digestibility of NDF. Compared to lambs fed with NLP, the MON group exhibited a 17% decrease in daily DM intake without impairing ADG. Depression in DM intake with MON supplementation is consistent with the higher ruminal propionate proportions, which have been reported to regulate feed DM intake through the liver oxidation process [35], resulting in more energy available for further metabolism [1]. Moreover, decreased nutrient intakes in lambs fed MON compared to those fed control and NLP diets may be due to the suppressing effect of MON on DM and OM digestibility [36]. Dietary supplementation with MON had a diminished digestive passage rate in the rumen and subsequently increased rumen fill, which is positively associated with decreased feed DM intake [37]. Adding NLP to a lamb’s diet exhibited greater CP intake, which corresponded well with increased daily DM intake [38], which might be due to changes in rumen microflora structure, resulting in increased nutrient digestibility [39] in comparison with the MON group.

Increased CP digestibility could be a result of the protein-binding ability of neem leaf phytochemicals formed with dietary CP [40], leading to more BW gain in comparison to the control. This finding agrees with Jack et al. [13], who reported increased CP digestibility owed to adding neem fruit to the ram’s diet. Compared to NLP, decreased OM and NDF digestibility in lambs fed with MON might be attributed to the inhibitory mechanism of MON on ruminal Gram-positive (fibrolytic degrading) rather than Gram-negative populations [3,41,42]. This finding is likely due to the reduction in ruminal pH, which correlated well with reduced acetate proportions in comparison with NLP. Similar to our results, Yang et al. [43] reported that the antiprotozoal potential of MON may be another reason behind reduced NDF digestibility. Increased NDF digestibility with lambs fed NLP coincided with results reported by Brooker et al. [44]. This may possibly be due to the presence of tannin-tolerating bacteria in the rumen. Furthermore, McSweeney et al. [45] found that ruminal fiber-degrading fungi have shown lower susceptibility to the suppressing effect of phenolic components than cellulolytic bacteria. 

The actuation mode of phytonutrient substances is mostly similar to that of ionophore additives, with no remaining final residues as reported by Coelho et al. [46]. However, the mode of action of MON in ruminant diets is well known to modify the rumen microbiome ecosystem [42]. Based on our data, NLP potential responses may refer to a different selective mechanism, which might have resulted in their differences in nutrient intake, fermentation, and digestion when compared to MON. Unfortunately, the NLP mechanism of action in this study has not yet been fully elucidated. The antimicrobial mechanism of phytogenic supplements can inhibit the growth of rumen Gram-positive (fiber-digesters) microflora while promoting the proliferation of Gram-negative (starch-digesters) bacteria [47]. Therefore, it seems that adding NLP to the diet effectively modified some specific ruminal Gram-positive microflora activities [48], thereby increasing OM digestibility in comparison to the MON group.

The binding properties of phytochemical constituents (either phenolics or flavonoids) present in neem leaves [18] might form complexes with dietary proteins [49], allowing greater fecal nitrogen excretion by lambs fed NLP than those fed MON. Based on this, the shifting nitrogen excretion dynamics from urine toward feces reveals a desirable benefit from an environmental perspective [9]. Furthermore, a significant secretion of endogenous fecal nitrogen could be another reason behind the increased fecal nitrogen excretion with the NLP diet. On the other hand, low fecal nitrogen excretion in lambs fed MON compared to NLP might indicate an efficient use of available nitrogen in the small intestine as a consequence of ruminal microbiota alterations [50]. Although lambs fed NLP showed higher CP intake than those fed MON, no differences were observed in CP digestibility or nitrogen balance between these supplements. The reason why both supplements did not modify nitrogen balance may be justified by the similar CP digestibility observed between lambs fed NLP and those fed MON.

### 4.3. Rumen Fermentation Variables

In recent years, the nutritional potential of phytonutrient components on ruminal fermentation, nutrient digestibility, and blood metabolites, which, accordingly, contribute to animal health and productivity, has been assessed [7,51,52]. The changes in ruminal pH are mostly associated with the changes in fermentation variables, including microbial activity and total VFA concentrations as reported by Allen [53]. In contrast, increased ruminal pH in lambs fed NLP was due mainly to NLP’s ability to effectively reduce total VFA concentration in the rumen when compared to MON. The opposite result (reduced ruminal pH and increased total VFA concentration) was observed with the use of MON, revealing that MON may affect the fermentation pattern through the alteration of microbial populations and/or their metabolic pathways [54]. This result agrees with previous findings observed with the use of MON [35,49].

The binding mechanism of phenolic compounds with dietary CP [40] has been shown to hinder CP degradability in the rumen, thereby increasing ruminal escape protein [55,56]. Reduced ruminal NH_3_-N concentration is related to the presence of phytochemical secondary compounds that are able to suppress the hyper-ammonia-producing bacteria population in the rumen [57]. Compared to the control group, reduced ruminal protozoan abundance in lambs fed NLP agrees with the findings of Patra et al. [43] and Yang et al. [4], who reported antiprotozoal activity of bioactive phenolic compounds in neem leaves by reducing the permeability of the protozoa cell membrane. However, the reduction in ruminal protozoal counts in MON is due to the known antiprotozoal effect of MON [58,59].

Despite the lack of effect on VFA components by NLP supplementation, the addition of NLP reduced total VFA concentrations in comparison to the control diet, indicating some inhibitory effects of NLP on rumen microbiota communities [40]. Previous studies have confirmed the discrepancies in the potential of phytochemical supplements on ruminal total and individual VFAs, which could be attributed to their plant sources, dosages, composition, experimental diets and conditions, and adaptation period as well [32]. Research studies have reported that the presence of a wide variety of bioactive substances (flavonoids and tannins) in neem leaves [60] may modify feed digestion [55], resulting in better nutrient utilization [61]. Increased ruminal acetate and decreased propionate proportions with NLP supplementation compared to MON are most likely attributed to rapid fiber degradation in the rumen, thereby increasing NDF digestibility. Whereas, the shift in ruminal VFA pathways in lambs fed MON from acetate to propionate was apparently because of diverting ruminal H_2_ production to propionate production, thereby reducing the acetate:propionate ratio [62]. Thus, MON was employed as a positive control. Reduced ruminal acetate proportions with MON were probably owing to the observed diminution in fiber digestibility, triggered by the antimicrobial properties of MON against ruminal cellulolytic-digesting bacteria [63].

### 4.4. Blood Metabolites and Fecal Egg Count

Changes in blood hematological and biochemical metabolites help to monitor and assess an animal’s physiological, nutritional, and pathological conditions [7]. Compared to the control group, there was no effect in response to both supplement groups on RBCs, Hb, and WBCs, except for the observed reduction in PCV percentage in lambs fed NLP and MON. Based on these results, lambs were under normal health conditions, as evidenced by the enhanced feed DM intake and BW gain of lambs. Differences in blood hematological and biochemical variables and enzyme activities in lambs fed with the NLP supplement compared to the control and MON groups were within the accepted normal range of healthy lambs previously reported for Omani sheep [64] under similar conditions.

The positive impact on nutritional and immunity status may be due to bioactive phenolic agents (such as tannins), which are capable of increasing dietary CP escape from the rumen to the abomasum and intestines [47]. Similar to our findings, increased blood total protein and globulin levels were attributed to the presence of phytonutrient compounds [65,66]. The increased blood globulin is similar to the report of Lakhani et al. [67], where a diet supplemented with phytogenic supplements rich in tannins indicated an efficient immune response in buffalo calves.

In the present study, reduced blood urea nitrogen concentration with NLP compared to the control was positively correlated with ruminal NH_3_-N concentration reduction [68,69], which is ascribed to the antimicrobial effects of phytochemical secondary metabolites [70]. According to Norouzi et al. [71], a decreased blood urea level was negatively correlated with animal BW gain. These results suggest that adding NLP to lambs’ diets implies efficient ruminal dietary CP utilization, allowing greater protein supply and proliferation for animals [33,72].

The changes in total bilirubin levels in the blood of lambs fed NLP compared to MON could be explained by the potential of polyphenolic metabolites present in herbal supplements on the gene responsible for glucuronidation of bilirubin that might modify blood bilirubin levels [73]. In this study, the presence of phytochemical bioactive constituents [73] in NLP resulted in an abnormal elevation of blood-liver enzymes compared to the control diet. Further, increases in liver endogenous enzyme secretion are directly associated with the physiological alterations in the liver caused by the presence of polyphenolic compounds in the diet [74]. Lakhani et al. [67] reported that dietary addition with phytogenic supplements (such as *Azadirachta indica*) had similar results for blood total protein, globulin, liver enzymes (GOT and GPT), and immune status, but no effects were observed in serum ALP value compared to the control in growing buffalo calves. Similarly, increased blood GPT concentrations demonstrate effective stress alleviation [67]. These results are contrary to the findings of Rivera-Chacon et al. [57], where blood ALP and GGT concentrations did not change with phytogenic supplementation in Holstein cows. Therefore, further research confirmations are still required to investigate the role of NLP in sheep health and immune status under long-term effects.

In our experiment, it was expected that dietary supplementation with NLP would decrease lambs’ total fecal egg count compared to those fed the control diet. However, the lack of differences in total FEC between lambs fed NLP and those fed control suggests that the concentrations of NLP phytochemicals used in this study might be less than those required to effectively reduce FEC. Whereas, decreased total FEC with MON supplementation was attributed to its ability to combat parasitic infections in livestock [75]. Despite these differences, lambs fed NLP showed neither clinical signs nor a significant growth delay compared to the control or MON, indicating that the amount of NLP used was safe for the lambs’ physiological and health status.

## 5. Conclusions

Under actual feeding conditions of our study, the addition of NLP additive is recommended as a novel natural feed supplement rather than MON to modulate the ruminal fermentation profile and enhance fibre digestibility. The addition of NLP increased growth performance with no adverse effects on lamb’s intake, digestibility of nutrients, and feed efficiency compared to the control although CP digestibility was improved. Furthermore, the results revealed that NLP with 40 g/kg DM could be used as a phytogenic feed supplement to promote body weight in growing lambs. However, further research confirmation is necessary to elucidate the potency of NLP on carcass quality and characteristics.

## Figures and Tables

**Table 1 animals-12-03146-t001:** Feed ingredient proportions and chemical composition of experimental diets on a dry matter basis (g/kg DM).

Item	Neem Leaves	Rhodes Grass Hay	Concentrate Mixture	Basal Diet ^1^
Ingredients, g/kg DM				
Rhodes grass hay	—	—	—	600
Yellow corn	—	—	541	216
Soybean meal	—	—	230	92.0
Wheat bran	—	—	206	82.4
Limestone	—	—	14	5.60
Sodium chloride	—	—	6.00	2.00
Mineral and vitamin mixture ^2^	—	—	3.00	2.00
Metabolizable energy ^3^, Mcal/kg DM	—	2.03	2.60	2.21
Chemical composition, g/kg DM				
Organic matter	875	910	884	902
Crude protein	90	87.0	162	109
Extract ether	35	17.5	33.0	22.0
Non-fiber carbohydrates ^4^	434	105	359	183
Neutral detergent fiber	316	700	329	589
Acid detergent fiber	273	476	109	366

^1^ Consisted of 600 g/kg roughage (Rhodes grass hay) and 400 g/kg concentrate feed mixture; ^2^ Mineral and vitamin mixture provided the following: Ca, 1.00%; P, 0.60%; Se, 0.3 mg/kg; vit A, 8800 IU/kg; vit D, 2200 IU/kg and vit E, 33 IU/kg. ^3^ Calculated according to NRC [19]. ^4^ Non-fiber carbohydrates (g/kg) = 1000 − (NDF + CP +EE + ash).

**Table 2 animals-12-03146-t002:** Influence of neem leaf powder or monensin supplementation on growth performance of Omani growing lambs.

Item	Diet *			SEM	*p*-Values
	Control	MON	NLP		
No. of Animals	6	6	6	—	—
Dry matter intake					
g/day	954 ^a^	810 ^b^	950 ^a^	12.6	<0.001
g/kg^0.75^	71.9 ^a^	59.5 ^b^	70.4 ^a^	1.21	0.002
Initial body weight, kg	22.6	23.0	22.6	0.75	0.961
Final body weight, kg	31.3 ^b^	32.5 ^a^	32.3 ^a^	0.80	0.029
Average daily gain, g/day	97.3 ^b^	107 ^a^	106 ^a^	0.01	0.014
Feed efficiency	0.106 ^b^	0.131 ^a^	0.113 ^b^	0.0064	0.002

^a,b^ Means on the same row with a different subscript are statistically different (*p* < 0.05). * Control: diet without supplements; MON: control diet plus 35 mg monensin/kg DM; NLP: control diet plus 40 g neem leaf powder/kg DM. Feed efficiency: average daily gain/dry matter intake; SEM: standard error of the mean.

**Table 3 animals-12-03146-t003:** Influence of neem leaf powder or monensin supplementation on nutrient intake, apparent digestibility, and nitrogen utilization of Omani growing lambs (collection period = 6 consecutive days).

Item	Diet *			SEM	*p*-Values
	Control	MON	NLP		
No. of animals	6	6	6	—	—
Nutrient intakes, g/day					
Dry matter	1048 ^a^	946 ^b^	1047 ^a^	12.3	0.014
Organic matter	943 ^a^	851 ^b^	942 ^a^	11.1	0.005
Crude protein	123 ^a^	111 ^b^	123 ^a^	0.3	0.034
Ether extract	24.8 ^a^	23.4 ^b^	24.8 ^a^	1.45	0.013
Neutral detergent fibre	578 ^a^	522 ^b^	578 ^a^	6.8	0.024
Acid detergent fibre	345 ^a^	311 ^b^	345 ^a^	4.1	0.043
Nutrients digestibility, g/kg					
Dry matter	694 ^a^	657 ^b^	682 ^a,b^	5.2	0.031
Organic matter	692 ^a^	655 ^b^	686 ^a^	4.2	0.026
Crude protein	543 ^b^	561 ^a^	655 ^a^	3.3	0.038
Ether extract	569	556	493	20.7	0.310
Neutral detergent fibre	685 ^a^	655 ^b^	692 ^a^	4.7	0.013
Acid detergent fibre	652	645	651	6.3	0.895
Nitrogen utilization, g/day					
Total nitrogen intake	19.6 ^a^	17.7 ^b^	19.6 ^a^	0.23	0.005
Total nitrogen excretion	15.6	14.3	14.7	0.45	0.493
Fecal nitrogen	11.3 ^a^	10.6 ^b^	11.5 ^a^	0.37	0.042
Urinary nitrogen	4.32	3.73	3.18	0.208	0.118
Retained nitrogen	3.96	3.34	4.86	0.398	0.321

^a,b^ Means on the same row with a different subscript are statistically different (*p* < 0.05). * Control: diet without supplements; MON: control diet plus 35 mg monensin/kg DM; NLP: control diet plus 40 g neem leaf powder/kg DM. SEM: standard error of the mean.

**Table 4 animals-12-03146-t004:** Influence of neem leaf powder or monensin supplementation on ruminal fermentation constituents of Omani growing lambs.

Item	Diet *			SEM	*p*-Values
	Control	MON	NLP		
No. of animals	6	6	6	—	—
pH	6.69 ^a^	6.49 ^b^	6.63 ^a^	0.082	<0.001
NH_3_-N, mg/dL	26.2 ^a^	25.0 ^b^	23.2 ^b^	0.24	<0.001
Protozoa, log10 cells/mL	5.56 ^a^	5.37 ^b^	5.05 ^b^	1.584	0.006
Total VFAs, mmol/L	98.4 ^a^	99.6 ^a^	77.8 ^b^	0.52	<0.001
Individual VFAs, % of total VFAs					
Acetate	52.9 ^a^	45.6 ^b^	52.4 ^a^	0.78	<0.001
Propionate	22.4 ^b^	28.2 ^a^	23.2 ^b^	0.34	0.017
Butyrate	15.6	15.0	14.8	0.08	0.561
Valerate	1.94	1.86	1.68	0.128	0.432
Isobutyrate	3.21	3.32	2.89	0.242	0.336
Isovalerate	6.39	6.14	5.51	0.067	0.345
C2:C3 ratio	2.43 ^a^	1.70 ^b^	2.31 ^a^	0.472	0.001

^a,b^ Means on the same row with a different subscript are statistically different (*p* < 0.05). * Control: diet without supplements; MON: control diet plus 35 mg monensin/kg DM; NLP: control diet plus 40 g neem leaf powder/kg DM. NH_3_-N: ammonia nitrogen; VFAs: volatile fatty acids concentration; % of total VFAs: molar proportions of individual VFAs; C2:C3: to propionate ratio; SEM: standard error of the mean.

**Table 5 animals-12-03146-t005:** Influence of neem leaf powder or monensin supplementation on blood parameters and fecal egg count of Omani growing lambs.

Item	Diet *			SEM	*p*-Value
	Control	MON	NLP		
No. of animals	6	6	6	—	—
Hematological parameters					
Red blood cells, ×10^6^/µL	12.1	12.0	11.8	0.06	0.218
White blood cells, ×10^3^/µL	7.78	8.31	8.37	0.253	0.586
Packed cell volume, %	39.8 ^a^	38.5 ^b^	38.0 ^b^	0.24	0.009
Hemoglobin, g/dL	14.2	13.8	13.9	0.11	0.306
Biochemical metabolites					
Total Protein, g/dL	6.18 ^b^	6.28 ^a,b^	6.43 ^a^	0.035	0.017
Albumin, g/dL	2.41	2.42	2.47	0.017	0.339
Globulin, g/dL	3.77 ^b^	3.86 ^a,b^	3.96 ^a^	0.036	0.042
Creatinine, mg/dL	1.25	1.34	1.34	0.017	0.071
Urea, mg/dL	15.3 ^a^	12.3 ^b^	12.6 ^b^	0.78	0.032
Glucose, mg/dL	93.5	92.1	95.2	1.64	0.742
Total bilirubin, mg/dL	0.124 ^a,b^	0.114 ^b^	0.143 ^a^	0.0011	0.021
Enzymes activities, IU/L					
Glutamic pyruvic transaminase	21.2 ^b^	18.2 ^c^	24.6 ^a^	0.47	<0.001
Glutamic oxaloacetic transaminase	103 ^b^	94 ^b^	131 ^a^	3.3	<0.001
Gamma-glutamyl transferase	42.7 ^b^	42.9 ^b^	49.1 ^a^	1.04	0.022
Alkaline phosphatase	962 ^a^	766 ^b^	930 ^a^	31.3	0.033
Fecal egg count ^1^, egg/g	3.02 ^a^	2.43 ^b^	3.06 ^a^	0.061	<0.001

^a,b^ Means on the same row with a different subscript are statistically different (*p* < 0.05). * Control: diet without supplements; MON: control diet plus 35 mg monensin/kg DM; NLP: control diet plus 40 g neem leaf powder/kg DM. ^1^ Fecal egg count transformed by log10 (X + 100); SEM: standard error of the mean.

## Data Availability

Not applicable.

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
