# Peer review of "Effects of Neem (*Azadirachta indica*) Leaf Powder Supplementation on Rumen Fermentation, Feed Intake, Apparent Digestibility and Performance in Omani Sheep"

_animals, 2022, doi:10.3390/ani12223146_

Round 1
Reviewer 1 Report
Animals-1956546 submitted by El-Zaiat et al- “Effects of neem (Azadirachta indica) leaf powder supplementation on rumen fermentation, feed intake, apparent digestibility and performance in Omani sheep” studied the potential of dietary addition of neem (Azadirachta indica) leaf powder (NLP) when compared to monensin (MON) on ruminal fermentation, blood variables, feed intake, digestibility, and performance of growing lamb using 18 Omani lambs that were equally divided into 3 treatments in CRD design for 90 days. The authors concluded that neem leaves powder could be used as a promising phytogenic supplement in sheep diets rather than the use of synthetic growth promoters. The article has scientific and practical impact of alternative AGPs however, I have major comments that might be considered:
1. In the introduction section, please emphasis on the added value/novelty of this article as there are some researches on the AGPs and monensin and phytogenic (neem (Azadirachta indica) leaf powder in the literature.
2. L-64, these references may be valuable
-Shehata, A.A.; Yalçın, S.; Latorre, J.D.; Basiouni, S.; Attia, Y.A.; El-Wahab, A.A.; Visscher, C.; El-Seedi, H.R.; Huber, C.; Hafez, M. H; et al. Probiotics, Prebiotics, and Phytogenic Substances for Optimizing Gut Health in Poultry. Microorganisms 2022, 10 (2), 395; https://doi.org/10.3390/microorganisms10020395
3. L 91, the experimental protocol and ethical approve # should be stated.
4. L 101, Plz give brief discerption of a preliminary in vitro dose-response study?
5. L 108, what is the form of the experimental diet?
6. L 131, Plz fix the parenthesis.
6. L 319, what it is the name of the bioactive compounds in NLP that have antiprotozoal activity? please declare?
7. L 346, what it is the implementing of acetate and propionate production due to NLP and monensin for milk and meat productions in lambs, please declare?
8. L 314, you said that Reduced ruminal NH3-N concentration in lambs fed NLP rather than those fed control could be attributed to the binding of phenolic compounds with dietary CP [32], while in L 348-352, U said that In comparison to the control, increased blood total protein and globulin levels in lambs fed with NLP suggest that NLP did not negatively influence nutritional status and immunity response in lambs, reflecting protein catabolism [43]. However, increased blood total protein levels were attributed to the bioactive compounds in herbal feed supplements [44,45]. How can NLP in the same time bind dietary CP and increased blood total protein and protein catabolsim, Plz declare?
9. L 348-361, rewrite after deep thought?
10. L 377-378, Plz provide the normal range of liver biomarkers in lambs, you should provide a reason for these increases instead for looking for acquisitions?
11. L 394-395, Plz express the change in growth performance as %.
12. L 397-399, you attributed the differences to the observed differences 399
in CP digestibility, in L 419-421, U denied this “no positive or negative effects were observed on nutrient intakes, digestibility, or nitrogen utilization of lambs fed NLP compared to control, how comes you explain different effects by the same reason? By the way Tannins are very well know to reduce protein and several amino acid digestibility?
13. L 500-501, does an increasing in the liver biomarkers leakage enzymes is good indictors of healthy animals?
14. Discussion needs to be rewritten and shorten with emphasis on important and economical findings
15. There is no references reflecting 2022, it should be for references updating
16. I recommend a major revision
Best wishes
Author Response
Attachted

Reviewer 2 Report
The whole writing quality of manuscript is well, I think. Some minor revision suggestions are listed below:
L131: the first bracket should be reversed.
L223: the 3 in the NH3-N should be subscript
L230: Should there be a significant difference in protozoa between CON and NLP in table 2? I suggest the author to check it carefully.
L293: the nutrient intakes in table 5 can be omitted except for the dry matter intake, I think. Furthermore, the crude protein digestibility should be checked because it is inconsistent with the result calculated according to the utilization of nitrogen in table 5.
Reviewer 3 Report
Observations to Manuscript
General observations:
Overall, the manuscript reads well; however, some sentences sound repetitive or wordy. Also, some paragraph of the materials and methods, and discussion are hard to follow/understand. I recommend a detailed re-reading to rephrase those sentences.
Throughout the manuscript, several scientific names and Latin expressions should be italicize. For example: L21, 25, 68, 78, etc. Also, I suggest standardizing the name "NLP" and either using "neem leaves powder" or "neem leaf powder". Finally, I recommend using a hyphen when three or more consecutive references are indicate; for example [1–3].
The abstract should be re-written in no more than 200 words and must be in a single paragraph.
I highly recommend to show the results and discussion following the order mentioned in the materials and methods. Moreover, all ruminal fermentation variables analyzed should be mentioned in the section “2.5 Rumen fermentation variables” of the materials and methods. (The determination of volatile fatty acids should be described in section 2.5).
The discussion should include further arguments between studies or results found in the literature. Also, findings of this study should be linked with the suggested pathways or implications of NLP supplementation on the different variables evaluated. Most of the findings of this study are attributed to the bioactive compounds of NLP, did the author characterize NLP? What bioactive compounds are found in this additive?
Ethical approval is required for studies involving animals. The protocol code and date of approval should be mentioned in the manuscript.
Specific observations
L14: This sentence sounds repetitive. Consider not using the word “additives” twice in one sentence.
L29, 84, 499: Consider inserting a comma to separate the elements.
L41: dray? Correct the spelling (dry).
L43-44: The phrase “had no effect on” sounds wordy.
L47: “40 g of /kg DM”?
L56: This sentence sounds repetitive. Consider not using the word “enhancing” twice in one sentence.
L60-66: I suggest to rephrase this paragraph. It is hard to read.
L64: “as well as the risks to the general public's health.” I recommended adding references that state such risks to the consumers’ health.
L71-76: Are there any publications that test NLP in other productive species? I suggest providing key examples or evidence to support the use of NLP in livestock.
L89: Please provide additional information: temperature, equipment used, and evidence of the method relied on.
L90: neem leaf powder (NLP) was already mentioned above. Please provide the size of the screen used in the mill.
L93: “mean ± standard deviation”, consider omitting this sentence.
L95: Better not to mention the duration of the study in this sentence.
L103: The phrase “Lambs were housed in individual pens” sounds repetitive and is already mentioned above.
L112: “The growing experimental period”. Better not to mention this way.
L113: instalment?
L131: Correct this phrase “)six consecutive days)”.
L134-137: Please provide a reference that support this protocol.
L138-140: Please described this technique with sufficient details or cited the established method.
L141: Please consider re-naming this subsection. The author mentions “ruminal fermentation variables”, but only ruminal pH and protozoa count are described.
L143: Does feeding is referred to the feeding trial?
L144: Drowned?
L144: “Custom designed”, consider adding a hyphen.
L145: “and repeated up to 3 times in order to avoid saliva contamination and collect representative samples”, sentence is not clear, consider paraphrasing.
L153-164: The methodology described for the blood chemicals constituents is not clear. Was the 10 mL blood sample separated for hematological and biochemical analyses (please organize the variables listed)? Was all the blood centrifuged? What type of BD vacutainer was used (red or lavender cap)? Have you considered using the EDTA tubes (anticoagulant) for hematological analyses? Were all the variables described analyzed from blood serum? The protocol used to determine enzymes activities is not described.
L161: Please change the “;” for a comma.
L169: “Then the sample was crushed and mixed”, I suggest rephrasing this sentence. The word “homogenized” seems to be more suitable here.
L170: Does “liquid” is referred to supernatant or liquid phase?
L174-177: “After recording ruminal pH, samples (5 mL of rumen fluid) were squeezed through four layers of cheesecloth in order to eliminate large feed particles. Rumen samples were centrifuged (SIGMA Laborzentrifugen, D-37520, Osterode am Harz, Germany) at 2000 × g for 15 min at 4°C and the supernatant was collected and frozen at -20°C before being analyzed for ruminal ammonia (NH3-N) concentration”. This paragraph seems to be more suitable in the subsection of “ruminal fermentation variables”. Also, provide a reference that support this protocol.
L175: Does “rumen samples” means rumen fluid samples?
L178, 181, 182, etc.: NH3-N? Used the correct chemical formula.
L184-194: Paragraph is hard to understand, this needed to be re-written. In addition, please cited the established method.
L206: “H2SO4”? Write the chemical formula correctly.
L209 (Statistical analysis): Please provide information of the statistical analysis of the “Apparent nutrients digestibility and nitrogen balance”; based on the text, the collection/measurement period was different. Has the author considered the initial weight of the animals as an adjustment factor in the statistical analysis of the productive performance variables?
L259: SEM: standard error of the mean.
L264: “However, lambs fed MON reduced daily DM intake expressed in g/day and g/kg 0.75 (P < 0.001; P = 0.002) compared to those fed NLP and control”. The sentence sounds incomplete.
L304: The author mentioned “there is little information available…” but no study was cited.
L306-308: The author needs to mention how NLP reduces VFA concentration in the rumen.
L352-355: How does tannins stimulate the immune status of lambs? Is there any mechanism suggested? Why type of tannins are found in NLP?
L384: Were the lambs infected with parasites or were exposed to parasites? Better not to mention this way if there is no evidence of parasitism infections.
L401: Which bacterial communities?
L402: Total bacteria count was not measured in this study. It is not proper to conclude this way.
L432: Which desirable ruminal modifications is the author discussing about?
L448: Please mention the changes in rumen microflora that are related to a higher nutrient digestibility?
479: Please describe which ruminal microflora are involved in OM fermentation.
L497-503: This sentences sounds more suitable for the discussion section than for the conclusions. Please review the idea. I recommend highlighting the importance/implications of these findings.
Round 2
Reviewer 1 Report
Thank you for good revision and hard work
Author Response
Thank you very much for your previous comments that helped us improve this manuscript.
Reviewer 3 Report
The authors made an effort to address all comments and suggestions in this revised version. Thanks.
Author Response
We thank you very much for the comments and suggestions that were valuable and very helpful for revising and improving our manuscript.